# EGCG, a Green Tea Catechin, as a Potential Therapeutic Agent for Symptomatic and Asymptomatic SARS-CoV-2 Infection

**DOI:** 10.3390/molecules26051200

**Published:** 2021-02-24

**Authors:** Mukesh Chourasia, Purushotham Reddy Koppula, Aruna Battu, Madhu M. Ouseph, Anil K. Singh

**Affiliations:** 1Center for Computational Biology and Bioinformatics, Amity Institute of Biotechnology, Amity University, Noida, Uttar Pradesh 201313, India; mchourasia@amity.edu; 2Regeneron Pharmaceuticals Inc., Tarrytown, NY 10591, USA; purushotham.koppula@regeneron.com; 3Independent Researcher, 4320 W Deska Dr., Spokane, WA 99202, USA; arunavatsa@gmail.com; 4Department of Pathology and Laboratory Medicine, Weill Cornell Medical College, New York, NY 10065, USA; mouseph@med.cornell.edu; 5Department of Pharmaceutical Sciences, Washington State University College of Pharmacy, Spokane, WA 99202, USA

**Keywords:** COVID-19, SARS-CoV-2, catechins, EGCG, ubiquitination, ISGylation, inflammation, rheumatoid arthritis, vaccines, PLPro

## Abstract

Coronavirus disease 2019 (COVID-19), caused by severe acute respiratory syndrome coronavirus 2 (SARS-CoV-2), has emerged to be the greatest threat to humanity in the modern world and has claimed nearly 2.2 million lives worldwide. The United States alone accounts for more than one fourth of 100 million COVID-19 cases across the globe. Although vaccination against SARS-CoV-2 has begun, its efficacy in preventing a new or repeat COVID-19 infection in immunized individuals is yet to be determined. Calls for repurposing of existing, approved, drugs that target the inflammatory condition in COVID-19 are growing. Our initial gene ontology analysis predicts a similarity between SARS-CoV-2 induced inflammatory and immune dysregulation and the pathophysiology of rheumatoid arthritis. Interestingly, many of the drugs related to rheumatoid arthritis have been found to be lifesaving and contribute to lower COVID-19 morbidity. We also performed in silico investigation of binding of epigallocatechin gallate (EGCG), a well-known catechin, and other catechins on viral proteins and identified papain-like protease protein (PLPro) as a binding partner. Catechins bind to the S1 ubiquitin-binding site of PLPro, which might inhibit its protease function and abrogate SARS-CoV-2 inhibitory function on ubiquitin proteasome system and interferon stimulated gene system. In the realms of addressing inflammation and how to effectively target SARS-CoV-2 mediated respiratory distress syndrome, we review in this article the available knowledge on the strategic placement of EGCG in curbing inflammatory signals and how it may serve as a broad spectrum therapeutic in asymptomatic and symptomatic COVID-19 patients.

## 1. Introduction

Coronavirus disease 2019 (COVID-19) also known as severe acute respiratory syndrome coronavirus 2 (SARS-CoV-2) is the reason for one of the worst pandemics of modern times which affected nearly every country in the world, and it has been recognized as one of the greatest threats to the humanity. As of now nearly 100 million people have been infected with the global death count around 2.2 million. The United States of America has been hit the hardest among all the countries with 25 million confirmed cases and 0.42 million deaths as of 25 January 2021 (https://coronavirus.jhu.edu/). Although recent advancements in scientific research on COVID-19 has led to the development of various vaccines, such as mRNA-based vaccines by Moderna and Pfizer/BioNTech [1,2] which pave a way for ending the pandemic, their long-term safety, efficacy and immunogenicity towards SARS-CoV-2 spike glycoprotein (S protein) is yet to be evaluated. Both the companies claim that their respective vaccines are ~95% efficient in providing immunity against SARS-CoV-2. However, the constantly evolving virus, leading to the emergence of new strains, might derail the outcome of ending the pandemic. A new COVID-19 strain from the United Kingdom, B.1.1.1 SARS-CoV-2 also known as 20I/501Y.V1, VOC 202012/01, or B.1.1.7 with many novel mutations, a South African variant of SARS-CoV-2 (known as 20H/501Y.V2 or B.1.351) with few mutations, and Brazil variants of SARS-CoV-2 (known as P.1) with 17 unique mutations including three at the receptor binding domain of the spike protein, may chip away the protection these vaccines provide and impede the vaccination programs [3]. Scientists around the world are yet to understand the pathobiology of COVID-19 for efficiently targeting patients’ clinical symptoms. A multi-level solidarity for eradication of the pandemic is required for development of new clinically efficient drugs or repurposing of existing drugs as well as vaccination programs by various governments around the world.

It has been one year into the pandemic and the cure is far from found. At this moment of despair, researchers are considering the idea of repurposing various translational and alternative medicines. Finding new alternative strategies to circumvent COVID-19 may be the best strategy at this moment of hypermutated COVID-19 virus. Since all new mutations in the virus have been found to be more infectious and overwrite the rules of immunology in humans, we need to focus on our age-old alternative medicines such as catechins. SARS-CoV-2 triggers acute inflammatory response in the lung microenvironment. Compelling evidence suggests that EGCG, a catechin, ameliorates acute lung injury via regulating inflammatory cytokines in animal models [4] and may have profound significance as a drug of choice in pre or post COVID-19 disease stages.

Catechins primarily found in green tea, *Camellia sinensis*, are (−)-epicatechin (EC), (−)-epigallocatechin (EGC) and EGCG [5]. EGCG is an omnipresent flavonoid in the tea plant that has many health benefits including protection against degenerative diseases [6] and antitumorigenic [7], antioxidative [8], hypoglycemic [9], antimutagenic [10], antimicrobial [11] antiviral [12] anti-inflammatory and immunomodulatory [13,14] properties. Among its many beneficial effects, antiviral properties can serve as a future therapeutic utility for drugging COVID-19. EGCG has been demonstrated to abrogate porcine reproductive and respiratory syndrome virus [15], Zika virus [16], Chikungunya virus [17], Dengue virus [18], influenza A virus H1N1 [19], human immunodeficiency virus [20] and Ebola virus [21]. Furthermore, Tallei et al. have shown through molecular docking studies that EGCG binds to SARS-CoV-2 S protein and main proteinase (M^pro^) and has the potential to inhibit the proteins [22]. These examples provide a compelling reason for targeting SARS-CoV-2 with EGCG. One of the key strengths of catechins is their power to bind multiple proteins that may have some immunomodulatory function. During COVID-19 infection, patients are overwhelmed by acute inflammation in the lung triggered by proinflammatory cytokines. EGCG or other catechins may very well be utilized as repurposed medicines in this scenario. There is a wealth of knowledge on EGCG and inflammation in the scientific literature. This review aims to discuss the systematic inflammatory state of COVID-19 patients, provide evidence which suggests EGCG can be a bona fide molecule for targeting those inflammatory pathways, emphasize how COVID-19 transcribed genes such as papain-like protease protein (PLPro) may be worth targeting, and address the key question: can EGCG be used as a therapeutic intervention for SARS-CoV-2?

## 2. Results

### 2.1. Inflammation Associated with COVID-19 Pathobiology and Possible Targeting by Catechins

#### 2.1.1. Inflammatory Response in COVID-19 Infection

Long et al. [23] assessed the immunological parameters of 37 asymptomatic and 37 symptomatic patients and concluded that 18 key cytokines dysregulated upon SARS-CoV-2 infection were found to be highly expressed in symptomatic individuals. We subjected all 18 inflammatory cytokines, tumor necrosis factor (TNF)-related apoptosis-inducing ligand (TRAIL), macrophage colony-stimulating factor (M-CSF), growth-regulated oncogene-α (GRO-α), granulocyte colony stimulating factor (G-CSF), interleukin-6 (IL-6), interleukin-2 (IL-2), β-nerve growth factor (β-NGF), interleukin-10 (IL-10), monocyte chemoattractant protein-1 (MCP-1), Skp1-cullin 1-F-box (SCF), interleukin-15 (IL-15), interleukin-8 (IL-8), interleukin-7 (IL-7), interferon gamma-induced protein 10 (IP-10), platelet-derived growth factor-BB (PDGF-BB), interferon-γ (IFN-γ), interleukin-18 (IL-18) and interleukin-2Rα (IL-2Rα), to gene ontology studies using metascape.org web-based prediction of pathway changes (Figure 1). 

We observed in our initial, preliminary analysis the involvement and dysregulation of several pathways such as cytokine-cytokine receptor interaction, IL-10 signaling, leukocyte proliferation, phosphoinositide 3-kinase (PI3K)-protein kinase B (Akt) signaling pathway, regulation of peptidyl tyrosine phosphorylation, mimicking Influenza A infection etc. (Figure 1A). Further, our gene ontology study also suggests the elevated levels of all 18 cytokines in human serum mimicked clinical signs of inflammatory and degenerative diseases with rheumatoid arthritis being a top candidate among other diseases such as Crohn’s disease, brain ischemia, dermatitis, and cancer (Figure 1B). Interestingly, the combined effect of various cytokines dysregulated in COVID-19 was recognized as rheumatoid arthritis in gene ontology findings highlighting the commonality between these two diseases. Although COVID-19 has no association whatsoever with rheumatoid arthritis manifestation, the clinical manifestations of COVID-19 mimic that of rheumatoid arthritis. This includes inflammation of lung in COVID-19 which is clinically similar to inflammation of synovial joints in rheumatoid arthritis, role of matrix modifying enzymes in bone destruction that is similar to SARS-CoV-2 infected lung microenvironment, massive cell infiltration mediated by inflammatory cytokines and hyper proliferation of fibroblasts leading to fibrosis in both the diseases [24]. Proinflammatory cytokines such as IL-6, IL-1β, tumor necrosis factor-α (TNF-α), etc. are central in the clinical manifestation of both the diseases. COVID-19 survivors face a great challenge in recovering from secondary complications (long-haul COVID) contributed by dysregulated pleural and alveolar region damage, organ failures related to kidney dysfunction, heart related issues, thrombosis, probable irreversible fibrosis in the lungs, and can take months or even years to recover back to normalcy [25]. 

#### 2.1.2. Targeting Individual Cytokines Might Not Be a Viable Option for COVID-19 Amelioration

One of the characteristics of COVID-19 is the presence of elevated IL-6 serum levels and low IFN response [26]. One of the early papers by Alessandro et al. [27] categorized COVID-19 symptomatic patients into 3 categories based on serum IL-6 levels: low (less than 10 pg/mL), medium (ranging between 10–65 pg/mL) and high (more than 90 pg/mL). Patients with the highest amounts of IL-6 had increased C-reactive proteins (CRP), the metalloproteinase inhibitor, tissue inhibitor of metalloproteinase1 (TIMP1), and cardiac tissue damage marker creatine kinase M (CKM). They also observed patients with increased amounts of IL-6 levels had enriched candidates for the Janus kinase-signal transducer and activator of transcription protein (JAK-STAT) pathways. Luo et al. [28] identified that serum levels of IL-6 and CD8^+^ T cell count predicted the morbidity of COVID-19 patients. Patients with serum levels of IL-6 more than 20 pg/mL and with less than 165 CD8^+^ T cells/µL are moribund in hospital settings. Several other reports were published characterizing IL-6 as being central to respiratory distress syndrome (RDS) associated with COVID-19. Recent clinical trials of tocilizumab (Actemra) in COVID-19 patients found it to be promising towards abrogating symptoms but did not improve the morbidity outcome [29,30] and results remain ambiguous in the final outcome. However, tolerability of anti-IL-6 receptor monoclonal antibodies in humans justified tocilizumab as the choice of drug in clinical settings and it is even now acceptable to administer tocilizumab to moribund patients as a last resort to save one’s life. Since COVID-19 is a disease with elevated levels of various proinflammatory cytokines it may not be wise to conclude that a single medicine would cure the symptoms of COVID-19. Baricitinib, a JAK1/JAK2 inhibitor has been found to be effective in COVID-19 treatment [31]. Baricitinib not only inhibits IL-6 mediated JAK-STAT signaling, it may have shown beneficial effects in abolishing SARS-CoV-2 by also abrogating the interaction of the viral spike protein with angiotensin I converting enzyme 2 (ACE2) receptors. Nevertheless, various other clinical trials with anti-IL-6 receptor antibodies [32] concluded that mortality is subject to virtually a high state of hyper-inflammation. Bozzi et al. [33] reported that IL-1 antagonists, such as anakinra, may have promising results in patients with high inflammation state towards survival and recovery from COVID-19. The authors of this study also concluded that treatment of COVID-19 patients with anakinra in combination with methylprednisolone may be beneficial for survival of patients on mechanical ventilators.

#### 2.1.3. EGCG Ameliorates Interleukin-1 (IL-1)-Induced IL-6 Expression

As mentioned previously, rheumatoid arthritis is also a disease of hyper-inflammation caused by the action of various proinflammatory cytokines in the synovial microenvironment. It is mainly driven by a complex interplay of synovial macrophages and synovial fibroblasts in the synovial milieu and is exacerbated by the overproduction of IL-1β [34,35]. Elevated IL-1β levels lead to persistent activation of mitogen-activated protein kinase (MAPK) pathway and increased production of IL-6, IL-8, MCP-1 systemically as well as in the synovial microenvironment. Our previous work [14,34,35] has successfully shown that targeting MAPK at TNF receptor-associated factor 6-mitogen-activated protein kinase 7 (TRAF6-TAK1) signalosome in rheumatoid arthritis synovial fibroblasts using EGCG effectively curbs the IL-1 induced IL-6 production. Likewise, by targeting macrophages using specific inhibitors for TAK1, such as 5z7 oxozeaenol, we successfully abrogated the production of various inflammatory cytokines including IL-6 and IL-1 from macrophages and clinically improved the severity of disease in preclinical rodent models. EGCG was found to be effective in impeding downstream signaling in the IL-6 driven JAK-STAT pathway as well as IL-1 driven MAPK pathway at various stages of signal transduction. EGCG inhibited the activation of JAK2 [36] as well as phosphorylation of STAT3 at Serine 727 [37] and phosphorylation of STAT3 Tyrosine 705 [38] (Figure 2).

Likewise, EGCG impeded the function of IL-1β [14] by inhibiting TRAF6 through curbing its autoactivation brought about by mono-ubiquitination and by inhibiting TAK1 phosphorylation in the kinase domain which translated to effective slowing down of signaling events and suppression of IL-6 and IL-8 production. We believe strongly that targeting IL-6 induced JAK-STAT pathway using EGCG makes the perfect case for drug of choice in SARS-CoV-2 infected individuals. Moreover, based on the fact that all the 18 known serum biomarkers of COVID-19 are immunomodulatory, and the fact that EGCG has previously been shown to be effective on all those pathways (Table 1), administering EGCG to symptomatic and asymptomatic COVID-19 patients may help resolve issues pertaining to virus replication, curbing cell death, and resolving cellular responsiveness to inflammatory triggers through its pleotropic effect on various pathways. 

In support of our proposed rationale, clinical trials are underway paving the way for making EGCG a drug of choice for COVID-19 prevention and treatment. As an example, the MELISA Institute Genomics and Proteomics Research SpA, in a multicenter, double-blinded, randomized, placebo-controlled clinical trial, are testing Previfenon, an EGCG oral tablet, as pre-exposure chemoprophylaxis to SARS-CoV-2 on health care workers directly exposed to clinical care, daily contact, or traffic of individuals with suspected for COVID-19 and are yet to conclude their findings about the efficacy of EGCG in preventing COVID-19 [55].

### 2.2. Initial Cellular Events of COVID-19 and the Role of PLPro

#### 2.2.1. Chronic Inflammatory Response and Cytokine Storm in COVID-19

One of the intriguing aspects of COVID-19 is the undetectability of the disease for a few days ranging between day 4 to day 14 of viral exposure thus making early pharmacologic intervention of the disease a challenge. Moreover, on many occasions COVID-19 remains undetected in asymptomatic patients and is the main cause of high transmission rates in the community. Asymptomatic patients are unaware of their disease status and may unknowingly spread the infection to healthy people in close surroundings [56]. SARS-CoV-2 infection is predicted to spread through droplets or aerosol from breath of asymptomatic or symptomatic individuals. Upon infection, the virus binds to alveolar type 1 (AT1) and alveolar type 2 (AT2) cells (AT cells) [24], macrophages as well as dendritic cells (DCs) of the pleural microenvironment [57]. Although the DC is one of the finest antigen-presenting cells, there is uncertainty about its responsiveness as antigen presenting cells in the case of SARS-CoV-2 infection. SARS-CoV-2 utilizes ACE2 and transmembrane serine protease 2 (TMPRSS2) receptors available on AT, DCs and macrophages to attach to the cells followed by delivering its genetic material, a single stranded sense RNA (+RNA), in the cytoplasmic compartment of the infected cells. Sense (+) single stranded RNA starts transcribing immediately and amplifies its early genes followed by its own genome amplification and packaging using host cellular machinery [58]. Viral infection of AT and DCs causes massive cell death due to pyroptosis [59]. Pyroptosis results in the release of damage associated molecular patterns (DAMPs) containing ATP, DNA, RNA and apoptosis-associated speck-like protein (ASC) oligomers which further triggers proinflammatory response in the neighboring cells, including endothelial, macrophage and epithelial cells [60]. This further increases the production of various other pro-inflammatory cytokines through activation of the inflammasome pathway in the resident surrounding cells such as, IL1-β and IFN-γ production from macrophages. Furthermore, IL-1β contributes to extensive production of IL-6, macrophage inflammatory protein 1α (MIP1α), MCP1 etc. from adjacent cells to recruit other cell-types related to immune responsiveness such as natural killer cells (NK cells), T cells and B cells. This uncontrolled inflammatory response also known as the cytokine storm is the reason for the clinically observed respiratory distress syndrome [24,61]. This combined accumulation of many inflammatory and immune related cells in the lung microenvironment may worsen the disease by triggering fluid accumulation in the alveolar exudate [58]. Space created by cell death in the alveolar lining is further filled by hyperproliferative lung fibroblasts to prevent further leakage in recovering patients, which mimics pannus formation by fibrosis in rheumatoid arthritis, and is responsible for complications and death in patients during the post-recovery stages, and possibly long-term pulmonary morbidity [62,63,64]. Since fibroblasts are not efficient cells for oxygen exchange, patients further suffer hypoxia-like situations leading to multiple organ failure. SARS-CoV-2 infected cell death by pyroptosis may not give dendritic cells a chance to act as antigen presenting cells for T and B cells to trigger the costimulatory activation or clonal selection process which may be further exacerbated by low antibody specific immune response or lack of memory cells in COVID-19 patients [65]. 

#### 2.2.2. Cellular Ubiquitination and ISGylation Processes in Infection

ISGylation is a very important step for initiating the host immune response against the virus [66]. ISGylation is the process of tagging interferon stimulated gene (ISG)15, a ubiquitin (UB)-like protein, to host cellular proteins and is analogous to E1-E2-E3 system of Ubiquitin Proteasome Signaling (UPS). ISG15 is an interferon (IFN)-induced protein and is required at the time of microbial infection of host cells. Likewise, the ubiquitin proteasome system also plays a pivotal role in microbial infection and is necessary for curbing viral replication. Ubiquitination of any protein in a Lys48-linked polyubiquitin (K48) manner destines it to degradation while Lys63-linked polyubiquitin (K63) format of ubiquitination provides stability to the proteins and retention of their biological activity [67]. Any altered initial response by the host may have consequences for the host to clear pathogens efficiently. Cells recognize any entry of pathogens through their TLR4-TLR7 receptors followed by activation of the caspase function of retinoic acid-inducible gene (RIG-I) upon ubiquitination by tripartite motif-containing protein 25 (TRIM25). Chang et al. have recently shown in HEK-293T and A549 cells that RIG-I ubiquitination leading to RIG-I activation [68] is an essential step for Interferon regulatory factor3-Interferon regulatory factor7 (IRF3-IRF7) heterodimer activation [68]. IRF3-IRF7 activation simultaneously is also achieved by stimulator of interferon genes (STING) [66]. Activated IRF3-IRF7 stimulates the production of early responsive genes such as IFN-α, IFN-β and ISG15 in host cells. IFN-α, IFN-β further stimulate JAK1-STAT1 signaling to make more copies of ISG15 and subsets of ISGs [66]. Being early responsive genes during viral infection, we infer that ISGs might provide stability and activity to their adaptor proteins listed by Perng et al. [66] such as STAT1, RIG-I, tumor susceptibility gene 101 protein (TSG101), protein kinase RNA-activated (PKR), charged multivesicular body protein 5 (CHMP5), IRF3, UB, beclin 1 (BECN1), JAK1, phospholipase Cγ1, mitogen-activated protein kinase 3 (ERK1), parkin, interferon-induced protein with tetratricopeptide repeats 1 (IFIT1), myxovirus resistance protein 1 (MxA), tumor protein 53 (p53), filamin B, proliferating cell nuclear antigen (PCNA), β-catenin, hypoxia-inducible factor 1-alpha (HIF1α), IQ motif containing GTPase activating protein 1 (IQGAP1), ubiquitin-conjugating enzyme E2 13 (UBC13), ubiquitin-conjugating enzyme H6 (UBCH6), protein phosphatase 2Cβ (PP2Cβ), tripartite motif containing 25 (TRIM25), eukaryotic translation initiation factor 4E family member 2 (4EHP), cyclin D1, fusion protein of PML nuclear body scaffold protein and retinoic acid α receptor (PML–RARα) and tumor protein 63 isoform (δnp63α) [66]. We performed a gene ontology study of these reported ISG adaptor proteins followed by gene enrichment analysis which identified antiviral mechanisms by ISGs and regulation of viral processes as top enriched pathways (Figure 3) corroborating ISG15′s commitment to antiviral activity.

#### 2.2.3. PLPro Targets Cellular Ubiquitination and ISGylation Processes

One of the main paradigms of SARS-CoV-2 infection is its dormancy or ability to remain undetected by the immune system cells. In order to avoid initial immunological detection, SARS-CoV-2 elegantly and systematically utilizes the host cellular machinery to its advantage (Figure 4). 

The moment SARS-CoV-2 sense + RNA strand enters the cells, it hijacks the host transcription machinery and transcribes its early gene containing Non-structural proteins (Nsp)1, Nsp2, Nsp3 and Nsp4 proteins as a proprotein. PLPro, a papain-like protease, is a product of Nsp3 auto-cleavage and it exerts its proteolytic cleavage function at Nsp1-Nsp2, Nsp2–Nsp3 and Nsp3-Nsp4 junctions of the proprotein. The fully functional Nsp1 protein, cleaved by PLPro from the proprotein, binds to host ribosome and suppresses host specific mRNA translation [69]. Binding of Nsp1 favors SARS-CoV-2 and results in further translation of + strand RNA docked on the host ribosome. This may be the turning point in host-pathogen interaction that establishes COVID-19. The binding of Nsp1 to host specific ribosomes prevents the endogenous host mRNA translation relative to SARS-CoV-2 RNA translation. Moreover, SARS-CoV-2 upon entry followed by its replication and assembly manages to steal the early cellular interferon responsiveness [70] and remains unnoticed for a long time in asymptomatic individuals by dysregulating host cellular responsiveness to ISGylation and endogenous ubiquitination and deubiquitination process by UPS. As discussed earlier, ISGs and the UPS system are one of the first line of defense systems of innate immunity [66,71]. PLPro exerts its protease function by recognizing LXGG motifs [72,73] on Nsp1-Nsp2, Nsp2-Nsp3 and Nsp3–Nsp4 junctions of the viral proprotein. SARS-CoV-2 PLPro has been shown to exert its proteolytic function on K48 specific polyubiquitination chains [74,75] and ISG15 specific chains [76] on host proteins. One of the unique characteristics of SARS PLPro is its ability to cleave ubiquitin in a di-ubiquitin manner in contrast to the Middle East respiratory syndrome (MERS)-related coronavirus, or EMC/2012 PLPro, which cleaves ubiquitin in mono-ubiquitin format [74] and might be responsible for the increased pathogenicity of SARS-CoV-2 when compared to MERS-related coronavirus. Further, findings from Shin et al. [77] identifies SARS-CoV-2 PLPro being more efficient in cleaving ISG15 vis a vis UB K48 subunit when compared to the earlier SARS-CoV PLPro. Their interactome analysis suggests that SARS-CoV-2 PLPro is highly enriched in ISG15 tagged proteins while SARS-CoV is highly enriched with ubiquitin bound proteins. These findings provide the rationale for SARS-CoV-2 being more powerful in suppressing the initial host viral response by abrogating IFN production compared to SARS-CoV. Furthermore, the authors also investigated GRL-0617, a ligand inhibitor for PLPro, in curbing virus release in cell supernatant in a dose-dependent manner and provided rationale for targeting COVID-19 by targeting PLPro. Despite being 83% homologous, PLPro mimics the role of de-ubiquitinating enzymes and de-ISGylation enzymes differently in these two coronaviruses warranting a more aggressive therapeutic approach for the elimination of SARS-CoV-2. Thus, PLPro seems like an attractive central target to be drugged as it is strategically important in regulating antiviral cellular dynamics. Klemn et al. [78] recently characterized the PLPro protein from SARS-CoV-2 and the possibility of it being targeted based on mutational analysis. Briefly, the authors assessed PLPro’s capability to bind ISG15 and K48 chain removal in an in vitro setting and identified that its binding towards ISG15 (S1 ubiquitin-binding site) as well as K48 (S2 ubiquitin-binding site) are different because of engagement of two distinct domains. Furthermore, they also observed that PLPro mutation of R166S/E167R can abrogate its de-ISGylation and de-ubiquitinase function. Other site-specific mutations also weakened PLPro’s de-ISGylation and deubiquitinase functions such as N156E and Y171R. This work provides the rationale for efficient targeting of PLPro by protein ligand screening in silico and in vitro for potential therapeutic applications in COVID-19. 

#### 2.2.4. Catechins May Counteract Inhibition of ISGylation/Ubiquitination by PLPro

In order to predict the binding of various catechins to PLPro, we downloaded PLPro protein structure from PDB databases. Docking with various catechins was performed as mentioned in the methods section (Figure 5). EGCG, (−)-epicatechingallate (ECG), (−)-gallocatechingallate (GCG), (−)-catechingallate (CG), (−)-gallocatechin (GC), (−)-catechin (C), (−)-epigallocatechin (EGC), and (−)-epicatechin (EC) showed a docking score of −8.601, −8.566, −7.865, −7.498, −6.337, −6.329, −6.318 and −6.128 Kcal/mole respectively. Since ECG did not show proper pose in the cavity due to steric hindrance, it was submitted for flexible docking where residues from the 6 Å radius of the ligand were kept flexible. EGCG, CG, GCG and ECG showed a similar pose in the ligand binding site of PLPro. EGCG and CG shared a common interaction with the binding site residues viz trihydroxyphenyl rings of both catechins show H-bonds with backbone carbonyl oxygen of L162 and G163 residues of PLPro, sidechains of D164 and R166 interacted through H-bonds with the hydroxyl group of dihydroxychroman, Y268 showed two cation-pi interactions with both trihydroxyphenyl rings of EGCG and trihydroxyphenyl and dihydroxyphenyl rings of CG (Figure 5C,D). Due to the presence of (2S,3R) stereocenters of the benzopyran ring in GCG as compared to (2R,3R) in ECG, the orientation of GCG and ECG was different in the binding cavity (Figure 5G,H), that assisted ECG to interact with G266 and Q269 through H-bonds that are also present in the co-crystal ligand. GC and EGC oriented almost identically in the cavity and showed H-bond interaction with the backbone carbonyl oxygen of G163, side chains of D164 and R166; trihydroxyphenyl rings of these ligands show cation-pi with Y268 (Figure 5E,F). C and EC did not show H-bonding with R166. Almost all the catechins have hydrophobic interactions with M208, P247, P248, Y264 and Y273 residues. These docking studies suggest that the Y268 and D164 that show cation-pi and H-bond interaction with all catechins play important role in binding as has also been demonstrated by Fu et al. [79]. Therefore, on the basis of binding energy and interaction, EGCG and ECG were found to have more binding affinity towards PLPro as compared to the other catechins and might be more effective in inhibiting PLPro activity.

## 3. Methods

### 3.1. Molecular Docking of Catechins on PLPro

The protein structure of SARS-CoV-2 PLPro was downloaded from the Protein Data Bank (PDB ID: 7CMD). The protein structure has been subjected to protein preparation wizard to add hydrogen, assign correct bond order and generate protonation state of titratable residues at physiological pH (7.4). The 20 Å grid around the co-crystallized molecule has been generated using receptor grid generation of Glide. All catechins were energy minimized using optimized potentials for liquid simulations extended (OPLSe) force field [80] then subjected to ligand preparation in the LigPrep. First, the 10 conformations of each catechins were generated using standard precision (SP) docking, then these conformations of each ligand were again submitted for extra precision (XP) docking of Glide [81]. The selection of the best pose was made on the basis of energy and interaction with the protein. All the calculations were performed using Schrodinger suit 2018-3.

### 3.2. Gene Ontology Studies

Gene ontology studies were performed on the Metascape.org website with standard settings.

### 3.3. Keywords Searched 

EGCG, catechins, PLPro, COVID-19, ubiquitination, ISGylation, etc. were used as key words alone or in combination in National Center for Biotechnology Information (NCBI) PubMed and/or Google Scholar. 

## 4. Conclusions

ISG15-tagged IRF3, JAK1 and STAT1 trigger a heightened IFN response for clearance of virus as an early event, and K48 specific degradation of IκBα followed by activation of NF-κB prepares the host cells for antigen presentation and initial immunomodulatory cytokine production, such as IL-6, which further is required by T- and B-cells for efficient co-stimulatory signal and clonal selection [82] that results in a well-organized immunological clearance of the virus. SARS-CoV-2 PLPro preferentially targets ISG15 to curb the initial host antiviral response, making the virus remain undetected for several days which in turn makes therapeutic intervention a challenge. PLPro targeting by mutational analysis as well as ligand-based analysis [79,80] provides the rationale for drugging COVID-19 at the PLPro inhibition step. Our preliminary analysis using catechins by the molecular docking approach identifies their advantageous effect, particularly EGCG and ECG, in inhibiting PLPro. Since SARS-CoV-2 PLPro R166 mutation completely abrogated its deubiquitinase and deISGylation activity, binding of catechins to R166 is predicted to be inhibitory for its biological function. Although there are emerging reports of cellular toxicity of high doses of EGCG in vitro and in the human body [83,84,85], the very low oral bioavailability of EGCG [83,86,87] nevertheless makes it an effective therapeutic agent when the appropriate dosage is determined. Not only have we addressed SARS-CoV-2 PLPro as a specific target of catechins, but we have also provided ample literature-based findings of EGCG targeting the proinflammatory IL-1β and IL-6 signaling pathways in this review. These anti-viral and anti-inflammatory properties of EGCG could prove to be effective as a prophylactic measure and in combination with the current parenteral vaccines might help prevent severe COVID-19 infections in the community. Moreover, catechins present in green tea extract hold much promise in curbing COVID-19 disease pathobiology owing to their multidimensional anti-inflammatory effects in humans and their potential as natural therapeutics with minimal side effects needs to be explored further to help humanity overcome the present pandemic crisis. 

## Figures and Tables

**Figure 1 molecules-26-01200-f001:**
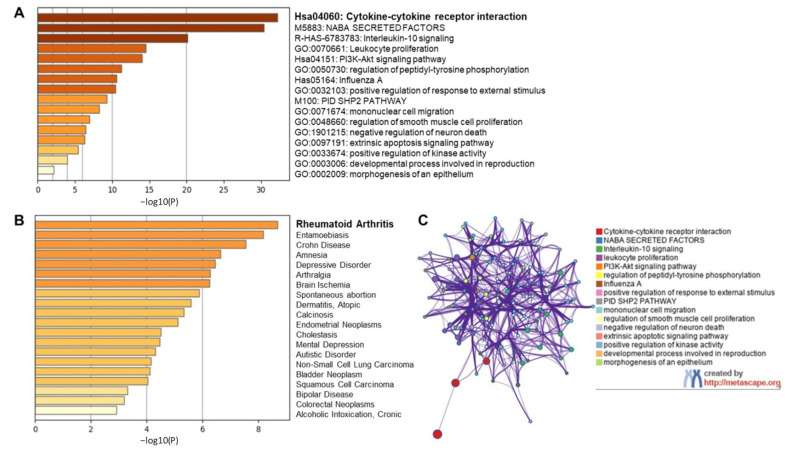
Gene ontology study of coronavirus disease 2019 (COVID-19) serum biomarkers. Gene ontology studies were performed on the 18 cytokines that are the signature biomarkers of COVID-19 infection, TRAIL, M-CSF, GRO-α, G-CSF, IL-6, IL-2, β-NGF, IL-10, MCP-1, SCF, IL-15, IL-8, IL-7, IP-10, PDGF-BB, IFN-γ, IL-18 and IL-2Rα, using metascape.org. (**A**) Top canonical pathways associated with COVID-19 inflammatory signature. (**B**) Top disease identifiers of inflammatory signature. (**C**) Network clusters associated with the canonical signature.

**Figure 2 molecules-26-01200-f002:**
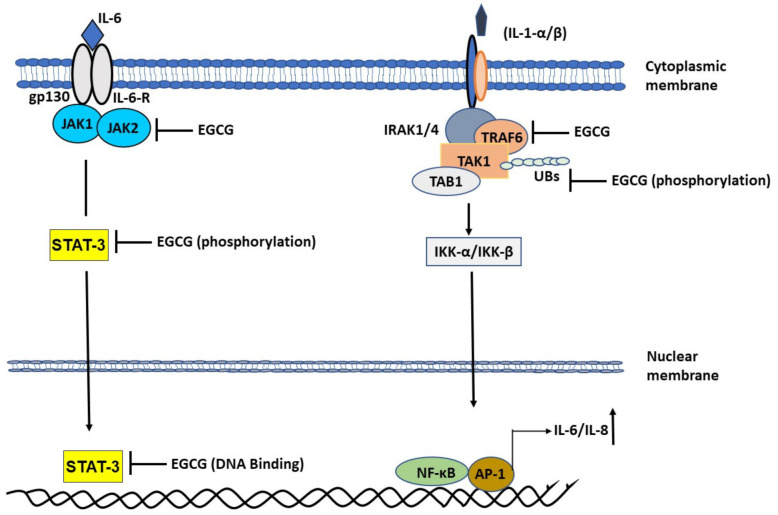
Role of epigallocatechin gallate (EGCG) in abrogating mitogen-activated protein kinase (MAPK) and Janus kinase-signal transducer and activator of transcription protein (JAK-STAT) pathway to attenuate the production of inflammatory cytokines. EGCG inhibits IL-1β induced TRAF6-TAK1 signal transduction complex formation. EGCG inhibits phosphorylation of Threonine 184/187 at its kinase domain. Inhibition of TAK1 abrogates NF-κB entry into the nucleus to abrogate IL-1 induced IL-6 and IL-8 production. EGCG also inhibits JAK2 activation and STAT3 activation by inhibiting Ser 727 and Tyr 705 phosphorylation, respectively. STAT3 binds to EGCG which suppress it DNA binding activity.

**Figure 3 molecules-26-01200-f003:**
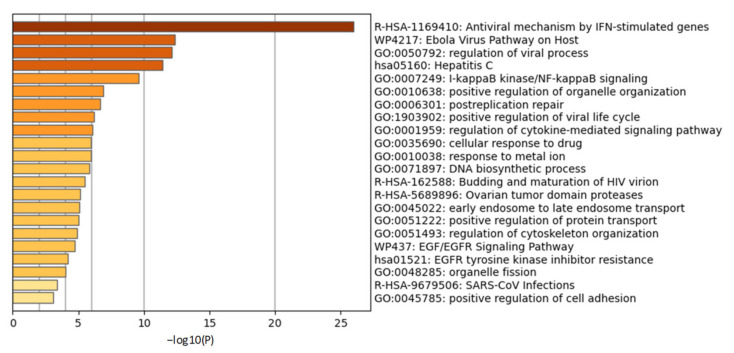
Gene ontology study of interferon stimulated gene (ISG) adaptor proteins. Gene ontology studies were performed on the reported ISG adaptor proteins STAT1, RIG-I, TSG101, PKR, CHMP5, IRF3, Ubiquitin, BECN1, JAK1, Cγ1, ERK1, Parkin, IFIT1, MxA, p53, Filamin B, PCNA, β-Catenin, HIF1α, IQGAP1, UBC13, UBCH6, PP2Cβ, TRIM25, 4EHP, Cyclin D1, PML–RARα and δnp63α using metascape.org. Gene enrichment analysis identified antiviral processes as top ontology terms.

**Figure 4 molecules-26-01200-f004:**
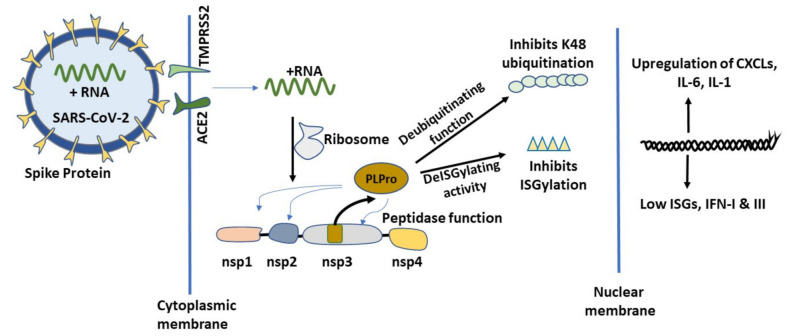
Role of severe acute respiratory syndrome coronavirus 2 (SARS-CoV-2) papain-like protease protein (PLPro) in COVID-19 infection. SARS-CoV-2 infects cells using host ACE2 and TMPRSS2 receptors. After infection, the virus uses the host transcription machinery and transcribes its early gene containing Non-structural protein (Nsp)1, Nsp2, Nsp3 and Nsp4 as a proprotein. PLPro, a papain like protease (PLP), is autocleaved from Nsp3 protein due to its intrinsic proteolytic function. SARS-CoV-2 PLPro then cleaves Nsp1–Nsp2, Nsp2–Nsp3 and Nsp3-Nsp4 junctions of the proprotein to give rise to the individual proteins. PLPro acts like a deubiquitinase and deISGylation enzyme on endogenous host proteins to abrogate host specific innate immune response.

**Figure 5 molecules-26-01200-f005:**
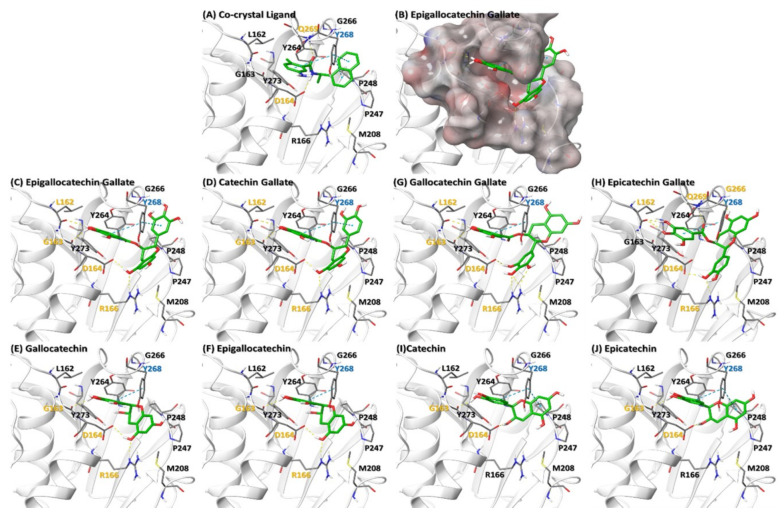
Binding of various catechins to SARS-CoV-2 PLPro protein. (**A**) Ligand (GRL0617) Co-crystal of PLPro (PDB ID: 7CMD). (**B**) Electrostatic surface of PLPro binding site (**C**–**J**) represent interaction of various catechins with the ligand binding site of PLPro. The residues in orange, cyan and black color represent H-bond, Cation-pi and hydrophobic interaction respectively. Orange dotted line represents H-bond whereas cyan dotted lines represent cation-pi interaction.

**Table 1 molecules-26-01200-t001:** Inhibitory effect of EGCG on the serum biomarkers of COVID-19.

Cytokine	Inhibitory Effect of EGCG	Reference
TRAIL	Suppresses TRAIL expression levels in auto immune thyroiditis.	[39]
	Sensitizes TRAIL induced apoptosis in ovarian cancer cells, via repression of nuclear factor kappa B (NF-κB) and inhibition of TNF-α and IL-6.	[40]
M-CSF	Down regulates M-CSF via miR-16 mediated inhibition of Tumor-associated macrophage (TAM) infiltration and M2 polarization in breast cancer cells.	[41]
GRO-α	Inhibits GRO-α expression levels by blocking IL-1β mediated stimulation in human Chondrocytes.	[42]
G-CSF	EGCG is chemopreventive through membrane type 1-matrix metalloproteinase (MT1-MMP) intracellular mediated signaling and sequential activation of the Src and JAK/STAT pathways thus antagonizing concavalin-A-induced colony stimulating factor (CSF)-2 and CSF-3 gene expression in mesenchymal stromal cells (MSCs).	[43]
IL-6	Inhibits IL-6 in epithelial ovarian cancer cells.	[44]
	Inhibits IL-6 expression levels by blocking IL-1β mediated stimulation in human chondrocytes.	[42]
IL-2	EGCG exerts anti-inflammatory effect by inhibiting macrophage migration inhibitory factor (MIF) in the pathogenesis of atopic dermatitis (AD).	[44]
	EGCG binds to staphylococcal enterotoxin B (SEB) and neutralizes it in a dose dependent manner and inhibits SEB-induced IL-2 in AD.	[45]
β-NGF	EGCG via the PI3K/Akt, glycogen synthase kinase-3 pathway and downstream signaling through cytochrome c and caspase-3 pathways could exert trophic factor effect in neurodegenerative diseases associated with oxidative injury.	[46]
	EGCG significantly exalted NGF-induced neurite outgrowth by increasing the expression levels of mRNA and proteins for the neuronal markers neurofilament-L and growth associated protein-43.	[47]
IL-10	EGCG ameliorated airway inflammation and eosinophil infiltrations in asthmatic mice by increasing the production of IL-10, the number of CD4^+^CD25^+^ Foxp3^+^ Treg cells and expression of Foxp3 mRNA in the lung tissue.	[48]
MCP-1	EGCG inhibits MCP-1, by suppressing IL-1β mediated stimulation in human Chondrocytes.	[42]
SCF	No articles found	
IL-15	EGCG regulates effector T cells and naïve T cell population and restores the balance of T helper (Th) cell 17/regulatory T cells, via STAT3 and STAT5. EGCG rescued IL‑7 production and decreased the levels of IL‑15 in transverse aortic constriction (TAC) rats with a therapeutic potential in inhibiting cardiac extracellular matrix remodeling.	[49]
IL-8	Inhibits IL-6 expression levels by blocking IL-1β mediated stimulation in human chondrocytes.	[42]
IL-7	EGCG suppresses IL-7 signaling by inhibiting the expression of IL-7R and IL-2R receptor subunit (common γ chain) involved in IL-2 mediated T-cell regulation.	[50]
IP-10	EGCG reduces airway inflammation via anti-inflammatory mechanism in the airway cells by binding directly to chemokines C-X-C ligand (CXCL)9, CXCL10, and CXCL11, thus limiting their biological activities.	[51]
PDGF-BB	EGCG, either by plasma membrane incorporated or soluble form, interacts directly with PDGF-BB via the galloyl group in the third position and interferes with PDGF-BB-induced mitogenic signaling pathway by inhibiting tyrosine phosphorylation of the PDGF-Rβ thereby preventing specific receptor binding, inhibiting downstream signal transduction pathways and cell proliferation.	[52]
IFN-γ	Reduced IFN-γ levels by suppressing NF-κB pathway.	[39]
	EGCG binds to staphylococcal enterotoxin B (SEB) and neutralizes it in a dose dependent manner and inhibits SEB-induced IFN-γ production and IL-2.	[45]
IL-18	EGCG attenuates microglial inflammation and neurotoxicity by suppressing NLRP3 and caspase-11-dependent inflammasome via toll-like receptor (TLR)4/NF-κB pathway in lipopolysaccahride+amyloid β protein (LPS+Aβ)-induced rat primary microglia and hippocampus of APP/PS1 mice by inhibiting the expression of ionised calcium-binding adapter molecule-1, cleaved IL-1β, and cleaved IL-18 induced by LPS+Aβ.	[53]
	EGCG exhibits anti-cancer and anti-atherosclerotic activity by a selective inhibition of the tyrosine phosphorylation of PDGF-Rβ and its downstream signaling pathway.	[54]
IL-2rγ	EGCG impacts T-cell regulation by inhibiting IL-2 proprietary α chains resulting in impaired signaling.	[50]

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
