# Peer review of "EGCG, a Green Tea Catechin, as a Potential Therapeutic Agent for Symptomatic and Asymptomatic SARS-CoV-2 Infection"

_molecules, 2021, doi:10.3390/molecules26051200_

Round 1
Reviewer 1 Report
This manuscript is remarkably interesting and full of detailed and useful notions for the reader. Only a few minor revisions:
Lines 216-219: In ref 56 (Koch, E. Previfenon® as chemoprophylaxis of covid-19 in health workers (herd)), a study on the prevention of respiratory disease caused by SARS-CoV-2 (COVID-19) in health care workers directly exposed to clinical care, daily contact, or traffic of individuals with suspected for COVID-19 during the epidemic outbreak was reported. On the contrary, the authors reported as follow: “MELISA Institute Genomics and Proteomics Research SpA are testing Previfenon, an EGCG oral tablet, on COVID-19 patients ………”. Please, clarify this point.
Lines 227-228: “COVID-19 infection is predicted to spread through droplets or aerosol…”. Remember that Covid-19 is the pathology, SARS-Cov2 is the etiological agent of Covid19 and the virus spread through droplets! Please, change the sentence.
Lines 238-259: references are lacking; please add.
Lines 261-283: the authors never specify whether these pathways have been seen activated in cell lines, primary cells or in vivo, or whether it is their own hypothesis.
Author Response
Reply to reviewers’ report (Reviewer 1)
Comment : This manuscript is remarkably interesting and full of detailed and useful notions for the reader. Only a few minor revisions:
Author’s reply: Thank you for finding our review worth. Reviewers' words are very encouraging to us. It will further motivate us to do better science.
Comment 1: “Lines 216-219: In ref 56 (Koch, E. Previfenon® as chemoprophylaxis of covid-19 in health workers (herd)), a study on the prevention of respiratory disease……”
Author’s reply: This was a mistake on our side and this sentence has been corrected to reflect the fact that the clinical trial intends to test the efficacy of Previfenon in the prevention of COVID-19 as a chemoprophylactic measure on health care workers directly exposed to clinical care, daily contact, or traffic of individuals with suspected for COVID-19. Please see Lines 260-266 of the revised manuscript.
Comment 2: “Lines 227-228: “COVID-19 infection is predicted to spread through droplets or aerosol…”. Remember that Covid-19 is the pathology…..”
Author’s reply: This sentence has been changed to ‘SARS-CoV-2 infection is predicted to spread….’ in line 275 of the revised manuscript.
Comment 3: “Lines 238-259: references are lacking; please add.”
Author’s reply: Pertinent references have been added to support the statements and are highlighted in yellow in lines 286, 290, 297, 299, 303 and 313 of the revised manuscript.
Comment 4: “Lines 261-283: the authors never specify whether…….”
Author’s reply: This section (3.2.2) summarizes the vast amount of literature on the ubiquitination and ISGylation processes and gives a general picture of how these processes function in the event of an infection. Since the focus of the manuscript is to explore the role of EGCG in modulating these processes, we have not included the details of the actual cell lines, etc. used to delineate these pathways. However, the relevant review articles as well as original articles that have been referred to in order to summarize these processes have been cited where appropriate (lines 316, 323, 329, 331, 333 and 335 of the revised manuscript). Since ISGs are early response genes in viral infection, we inferred that ISGs provide stability and activity to their adaptor proteins. Line 334 of the revised manuscript has been changed to portray this. Additionally, we performed a gene ontology study of the various ISG adaptor proteins and found them to be clustered into viral response pathways which corroborates our inference. These results have been included in the revised manuscript as a new Figure 3 and described in lines 346-371.
Reviewer 2 Report
In this manuscript, authors try provide evidence which suggests EGCG can be a beneficial molecule for targeting inflammatory pathways in cytokines storm resulting from COVID-19 infection. They emphasize that COVID-19 transcribed genes such as PLPro may be worth targeting. They also consider if EGCG can be used as a therapeutic intervention for SARS-CoV-2, as a molecule which interact with SARS-Co-2 PLPro enzyme and inhibit further process of viral infection.
The manuscript is very interesting, it provide many information about inflammation associated with COVID-19 pathobiology and possible targeting by catechins. There is also insight in initial cellular events of COVID-19 and the role of PLPro. Almost all manuscript is the overeview (line 87) of literature data concerning those topics. Finally, at the end of their work there are the molecular docking results of the binding of various catechins to PLPro. These are very important results but there are also known other proteins which are able to interact with EGCG (e.g. Spike Glycoprotein of SARS-CoV-2: Tallei TE, Scientifica (Cairo). 2020 Dec 23;2020:6307457. doi: 10.1155/2020/6307457. eCollection 2020.), as was also made by molecular docking. But authors do not mention about it. The should also indicate any other possible interaction of EGCG because it is possible that this compound and other catechins can interact with human proteins what would not be beneficial. Was there any microarray experiment performed to check this possibility?
Additional minor comment is that is mess with abbreviations. Sometimes they appear when the new name appears the first time, sometimes it never appear and sometimes it appear later. Because the article is rich in the short names of different compounds and proteins, the best would be the additional paragraph with the explanation of the abbreviations only.
Author Response
Comment 1: In this manuscript, authors try provide evidence which suggests EGCG can be a beneficial molecule for targeting inflammatory pathways…..
Author’s reply: Thank you for the detailed read and encouraging words. We strongly believe PLPro should be utilized as a bonafide new therapeutic target.
Comment 2: “These are very important results but there are also known other proteins which are able to interact……..”
Author’s reply: The interactions of EGCG with other SARS-CoV-2 viral proteins has been included in the revised manuscript in lines 95-97. However, we did not focus on the interaction of EGCG with the S protein since the spike protein has been mutating, giving rise to different strains of SARS-CoV-2, and might not be an effective druggable target in the long term.
Comment 3: “The should also indicate any other possible interaction of EGCG because it is possible that this compound……..”
Author’s reply: This has been addressed in lines 520-523 of the revised manuscript. Although EGCG has been reported to have adverse effects at high doses, the low oral bioavailabity of EGCG makes it a potential therapeutic to be explored for COVID-19 amelioration.
Comment 4: “Was there any microarray experiment performed to check this possibility?”
Author’s reply: We have not performed any microarray experiments to check the possibility of how EGCG interacts with other human proteins adversely.
Comment 5: “Additional minor comment is that is mess with abbreviations. Sometimes…..”
Author’s reply: We have included the abbreviations when the new names appear for the first time in the entire revised manuscript. Additionally, we have included a new Section 5 in the revised manuscript with all the abbreviations in order of their appearance as suggested by the reviewer.